# Inhibition of hepatic lipogenesis enhances liver tumorigenesis by increasing antioxidant defence and promoting cell survival

Marin E. Nelson[1], Sujoy Lahiri[1], Jenny D.Y. Chow[1], Frances L. Byrne[1,2], Stefan R. Hargett[1], David S. Breen[1], Ellen M. Olzomer[2], Lindsay E. Wu[2], Gregory J. Cooney[3,4,5], Nigel Turner[2], David E. James[3,4,5], Jill K. Slack-Davis[6], Carolin Lackner[7], Stephen H. Caldwell[8,9] & Kyle L. Hoehn[1,2,8,9]

The metabolic pathway of *de novo* lipogenesis is frequently upregulated in human liver tumours, and its upregulation is associated with poor prognosis. Blocking lipogenesis in cultured liver cancer cells is sufficient to decrease cell viability; however, it is not known whether blocking lipogenesis *in vivo* can prevent liver tumorigenesis. Herein, we inhibit hepatic lipogenesis in mice by liver-specific knockout of acetyl-CoA carboxylase (ACC) genes and treat the mice with the hepatocellular carcinogen diethylnitrosamine (DEN). Unexpectedly, mice lacking hepatic lipogenesis have a twofold increase in tumour incidence and multiplicity compared to controls. Metabolomics analysis of ACC-deficient liver identifies a marked increase in antioxidants including NADPH and reduced glutathione. Importantly, supplementing primary wild-type hepatocytes with glutathione precursors improves cell survival following DEN treatment to a level indistinguishable from ACC-deficient primary hepatocytes. This study shows that lipogenesis is dispensable for liver tumorigenesis in mice treated with DEN, and identifies an important role for ACC enzymes in redox regulation and cell survival.

[1] Department of Pharmacology, University of Virginia, Charlottesville, Virginia 22908, USA. [2] School of Biotechnology and Biomolecular Sciences, University of New South Wales, Sydney, New South Wales 2052, Australia. [3] Charles Perkins Centre, The University of Sydney, Sydney, New South Wales 2006, Australia. [4] School of Life and Environmental Sciences, The University of Sydney, Sydney, New South Wales 2006, Australia. [5] School of Medicine, The University of Sydney, Sydney, New South Wales 2006, Australia. [6] Department of Microbiology, Immunology and Cancer Biology, University of Virginia, Charlottesville, Virginia 22908, USA. [7] Institute of Pathology, Medical University of Graz, 8010 Graz, Austria. [8] Department of Medicine, University of Virginia, Charlottesville, Virginia 22908, USA. [9] Emily Couric Clinical Cancer Center, University of Virginia, Charlottesville, Virginia 22908, USA. Correspondence and requests for materials should be addressed to K.L.H. (email: k.hoehn@unsw.edu.au).

Primary liver cancer has a high mortality-to-incidence ratio of 0.93 and is the third leading cause of cancer-related death worldwide[1]. The most common histologic type of primary liver cancer is hepatocellular carcinoma (HCC). HCC has a high fatality rate because only ~30% of HCC patients are eligible for surgical resection or transplantation, and HCC is extremely resistant to chemotherapy[2]. The only Food and Drug Administration-approved targeted therapy for unresectable HCC is Sorafenib, a kinase inhibitor that is effective in a minority of patients and has survival benefits of ~12 weeks[3,4]. For many patients, palliative care is the best option[5]. New strategies for the prevention and treatment of HCC are needed.

The de novo lipogenesis pathway has received considerable attention as a therapeutic target for the treatment of liver cancer because it is frequently upregulated in liver tumour tissue and has prognostic significance for patient survival[6–9]. Lipogenesis is the metabolic pathway that converts amino acid and carbohydrate metabolites into lipids[10]. Lipids produced via lipogenesis have biochemical properties that are distinct from lipids extracted from the extracellular space, and several theories exist for why lipogenesis is upregulated in tumours. For example, lipogenesis-derived lipids are highly saturated and resistant to lipid peroxidation, saturated lipids may facilitate lipid raft formation to drive aberrant signal transduction pathways, de novo synthesized lipids may serve as endogenous ligands for transcription factors, and/or tumour cells may not be capable of scavenging enough extracellular lipids to support the increased rate of proliferation[11–15]. Thus, it is currently unknown whether lipogenesis-derived lipids are required for tumorigenesis, or whether circulating lipids are sufficient to satisfy the metabolic and physical properties required for tumour cell growth and proliferation.

The first rate-limiting step in de novo lipogenesis is catalysed by acetyl-CoA carboxylase (ACC) enzymes ACC1 and ACC2. Each ACC isoform is transcribed from a separate gene in both mice and humans. ACC1 converts acetyl-CoA to malonyl-CoA in the cytosol where malonyl-CoA is used as a substrate for lipid synthesis. ACC1 expression is frequently upregulated in human liver tumours and higher ACC1 expression correlates with worse prognosis[9]. ACC2 catalyses the same reaction as ACC1; however, ACC2 is localized to the cytosolic surface of mitochondria where it produces malonyl-CoA that inhibits carnitine palmitoyltransferase 1 (CPT1) to decrease fat oxidation. Despite their distinct compartmentalization, ACC2 can compensate for the loss of ACC1 and drive lipogenesis in ACC1-deficient hepatocytes[16]. However, deletion of both ACC1 and ACC2 in hepatocytes is sufficient to completely block lipogenesis[17].

ACC enzymes are considered to be potential drug targets for the treatment of liver cancer because knockdown of ACC1 in human HCC cells in vitro is sufficient to reduce cancer cell viability[9], while loss of ACC activity in the liver is not associated with deleterious consequences to normal non-cancerous tissues[17]. However, the role of ACC enzymes in endogenous liver tumorigenesis has not been functionally tested in vivo. Herein, the role of ACC enzymes in liver tumorigenesis was investigated by testing whether mice lacking hepatocyte ACC activity were susceptible to liver tumorigenesis caused by the hepatocellular carcinogen diethylnitrosamine (DEN). Surprisingly, mice lacking hepatocyte ACC expression had greater susceptibility to DEN-induced liver tumorigenesis. Mechanistically, blocking ACC activity conserved NADPH that would otherwise be used for lipogenesis, resulting in greater antioxidant defence that protected hepatocytes from oxidant-mediated cell death.

## Results

### Human and mouse liver tumours have a lipogenic phenotype.
The most common mouse model of liver tumorigenesis involves a one-time exposure to the hepatocellular procarcinogen DEN. DEN-induced tumours have genetic signatures related to poor prognosis HCC in humans[18], and the DEN model demonstrates similarities to human HCC in the context of gender bias, increased incidence with poor diet and the mechanism of tumour initiation involves genotoxic damage[19–23]. We first investigated whether the mouse model of DEN-induced liver tumorigenesis demonstrated a lipogenic phenotype that was comparable to human HCC tumours. ACC expression was evaluated in >300 human HCC and non-cancerous liver specimens available in Oncomine[24]. These data showed that ACC1 gene (ACACA) expression was twofold greater in HCC compared to non-cancerous liver (Fig. 1a). Of the HCC cases that had associated clinical data, ACACA expression positively associated with tumour grade (Fig. 1b). In contrast, ACC2 gene (ACACB) expression was decreased by 25% in HCC tissue compared to non-cancerous liver tissue and was not associated with tumour grade (Fig. 1c,d). Analysis of ACC expression in liver tissue of mice treated with DEN at 2 weeks of age and killed at 32 weeks of age revealed a twofold increase in Acaca expression and a twofold decrease in Acacb expression in tumours compared to non-tumour tissue (Fig. 1e,f). Thus, DEN-induced liver tumours closely resembled human HCC with respect to relative changes in ACC gene expression.

We next investigated the lipogenic phenotype of tumour-derived and non-tumour-derived liver cells of mouse and human origin. Rates of lipogenesis were measured in human HCC cells (HepG2 and HuH7), murine hepatoma cells (Hepa1-6 and Hepa-1c1c7) and non-cancerous hepatocytes including primary murine hepatocytes and human non-cancerous liver cells (PH5CH8). Compared to non-tumour-derived cells, all tumour-derived cells had significantly higher rates of lipogenesis irrespective of whether they were derived from mouse or human tissue (Fig. 1g). In contrast, no cancer-specific alterations were observed in other parameters of cellular bioenergetics including basal oxygen consumption, extracellular acidification or spare respiratory capacity (Supplementary Fig. 1a–f). Thus, the lipogenic phenotype of liver tumour tissue was conserved between mouse and human tissues, and cells.

### Inhibition of hepatic lipogenesis promotes tumorigenesis.
To determine whether inhibition of hepatic ACC activity affected liver tumorigenesis, liver-specific ACC1/ACC2 double-knockout (LDKO) mice and wild-type floxed littermate controls (Flox) were exposed to DEN at 2 weeks of age and evaluated for tumours at 40 weeks of age. Both ACC isoforms were knocked out because ACC2 can compensate for lipogenesis in the absence of ACC1. Unexpectedly, we observed that mice lacking hepatic ACC activity had increased tumour incidence and multiplicity compared to Flox controls (Fig. 2a–c). Importantly, tumour cells derived from LDKO mice completely lacked ACC expression and were incapable of lipogenesis (Fig. 2d,e), indicating that tumours did not arise from non-hepatocyte cell types.

Metabolic disorders including obesity, hepatic steatosis, glucose intolerance and hyperinsulinaemia are associated with increased liver tumorigenesis in both mice and humans; however, none of these physiologic parameters were different between LDKO and Flox control mice (Fig. 3a–g). Because hyperproliferation is a common phenotype in hepatocyte transformation[25], we evaluated three markers of proliferation in LDKO and Flox liver by measuring hyperplastic foci by histology (Fig. 4a,b), alpha-fetoprotein (Afp) messenger RNA (mRNA) expression (Fig. 4c) and immunohistochemical staining for the proliferation marker Ki67 (Fig. 4d,e). Compared to Flox control mice, LDKO mice had evidence of hyperproliferation in all three assays. In contrast,

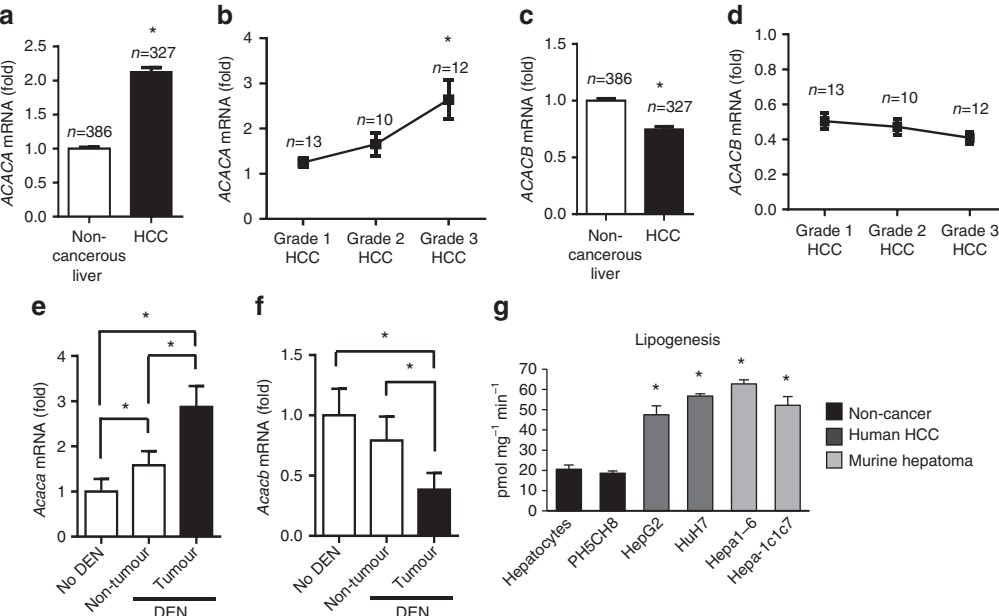

**Figure 1 | Lipogenic characteristics of human and mouse liver tumours.** (**a**) *ACACA* mRNA expression in human non-cancerous liver and HCC tumour tissue. (**b**) *ACACA* expression adjusted by HCC grade. For **b**, * indicates significant difference from Grade 1 HCC. (**c**) *ACACB* mRNA expression in human non-cancerous liver and HCC. (**d**) *ACACB* expression adjusted by HCC grade. (**e**) *Acaca* and (**f**) *Acacb* mRNA expression in liver non-tumour or tumour tissue of wild-type C57BL/6 mice untreated or treated with DEN at 2 weeks of age and collected at 32 weeks of age ($n = 4$ untreated and 9 DEN-treated mice). (**g**) Rates of lipogenesis from glucose in non-cancerous cells (primary murine hepatocytes and human PH5CH8 cells), and liver cancer cells lines from human (HepG2 and HuH7) and murine (Hepa1-6 and Hepa-1c1c7) origin ($n =$ at least three independent experiments). * Indicates significant difference, $P < 0.05$ as determined by two-tailed $t$-test (**b**,**d** and **e**–**g** were analysed by one-way analysis of variance followed by Tukey's *post hoc* analysis.) Data are represented as mean ± s.e.m.

cleaved caspase 3 (CASP3), a marker of apoptosis, was not significantly different between LDKO and Flox control liver (Supplementary Fig. 2a,b). Furthermore, because liver damage and inflammation can contribute to liver tumour development, we investigated genotype-specific differences in these parameters. Histopathologic analysis of liver tissue from 40-week-old mice revealed that neither genotype had evidence of liver damage in the context of hepatocellular ballooning, and analysis of portal and lobular inflammation showed similar numbers of inflammatory foci consisting mainly of mononuclear cells with occasionally admixed neutrophils (Fig. 5a,b; Supplementary Fig. 2c).

**Inhibition of lipogenesis alters hepatic lipid species.** To gain a better understanding of the metabolic changes caused by inhibition of lipogenesis, global metabolomics analyses were performed on non-tumour liver tissue from DEN-treated LDKO and Flox control mice (Supplementary Data 1). In light of the role of the ACC enzymes in lipid synthesis, we surveyed the metabolomics data set for ACC-dependent changes in lipid species (Supplementary Fig. 3a–g). We first investigated palmitate because it is a major end product of lipogenesis. Curiously, LDKO liver tissue had no change in palmitate (16:0), 16:0 ceramide (palmitoyl-sphingosine) nor in palmitate-based monoacyl-glycerols (MAGs) or phospholipids. These data suggest that palmitate is being provided by the diet or other tissues such as adipose (Supplementary Fig. 3a,b). However, not all lipid species were unaltered. LDKO liver tissue had ∼30% lower levels of stearate (18:0; Supplementary Fig. 3a), and stearate-based MAGs and sphingolipids were also decreased (Supplementary Fig. 3e,g). LDKO liver also had significantly higher levels of the poly-unsaturated fatty acids (PUFAs) linoleate (18:2) and linolenate (18:3; Supplementary Fig. 3a). Similarly, phospholipids with an 18:2 fatty acid at the carbon-2 position were significantly higher,

18:2 MAGs trended higher and 18:3 MAGs were significantly higher in LDKO livers (Supplementary Fig. 3c,e). Finally, LDKO livers had significantly lower levels of the PUFA arachidonate (20:4; Supplementary Fig. 3a) and lower phospholipids containing a 20:4 fatty acid at the carbon-2 position (Supplementary Fig. 3c). Taken together, levels of specific phospholipids, glycerolipids and sphingolipids generally corresponded with levels of the respective fatty acid components. The mechanism linking ACC activity to these changes in lipid species requires further elucidation, but one possibility is that some of the changes may be secondary to increased antioxidant status of the cell that would lessen the peroxidation of some PUFA species.

Because LDKO hepatocytes do not manufacture lipids *de novo*, it is likely that they extract more lipids from the circulation than Flox controls. To investigate this possibility, we measured the expression of lipoprotein lipase (LPL) and lipid transporters CD36 and FATP5 from LDKO and Flox liver tissues. LPL has been shown to be important for liver tumorigenesis by increasing exogenous lipid uptake[26]; however, we did not observe any change in LPL protein or mRNA expression (Supplementary Fig. 4a–c, uncropped blots are shown in Supplementary Fig. 5). In contrast, LDKO liver tissue had increased mRNA expression of the fatty acid transporters CD36 and FATP5 (Supplementary Fig. 4d,e). These data suggest that ACC-deficient hepatocytes have increased capacity to scavenge fatty acids.

**ACC activity affects antioxidant status and cell survival.** We next assessed whether ACC-deficient liver tissue had other alterations in global metabolites. Pathway enrichment analysis of metabolomics data revealed marked changes in glutathione antioxidant defence (Fig. 5c) that were consistent with increased products of the pentose phosphate pathway (PPP) including 6-phosphogluconate and NADPH (Fig. 5d,e). NADPH is a

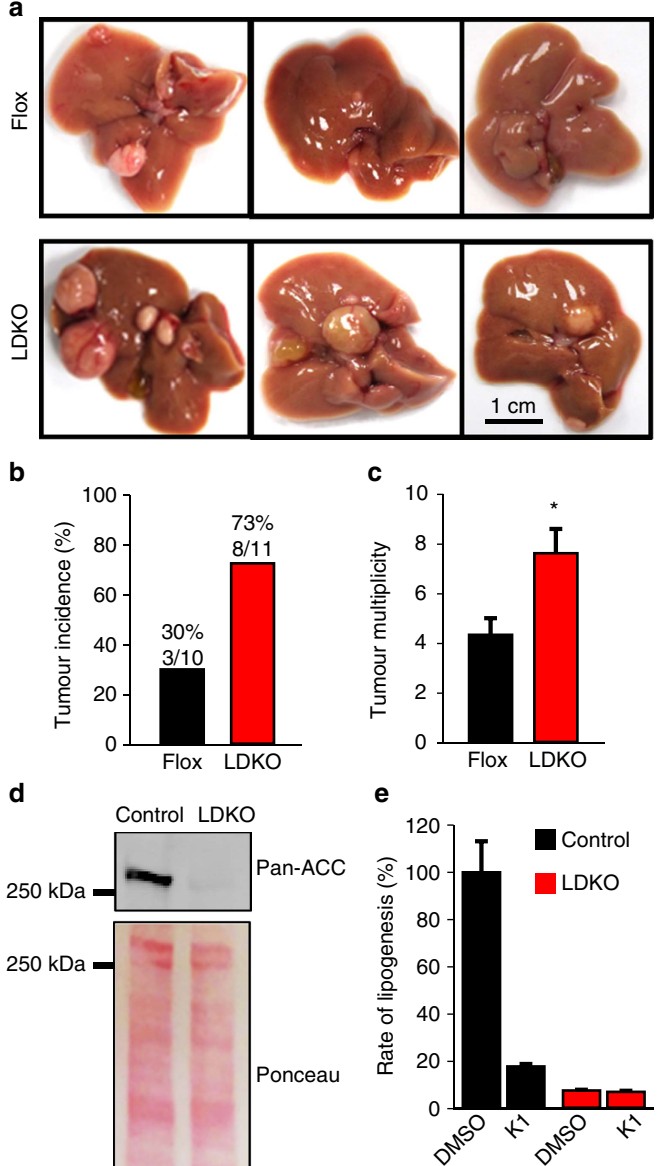

**Figure 2 | ACC deletion increases diethylnitrosamine-induced tumorigenesis in mice.** (**a**) Representative images of livers from LDKO and floxed littermate control (Flox) mice treated with DEN at 2 weeks of age and collected at 40 weeks of age. Scale bar, 1 cm. (**b**) Liver tumour incidence and (**c**) multiplicity of DEN-treated LDKO and Flox mice. (**d**) ACC protein expression and (**e**) rates of lipogenesis from acetate with or without the pan-ACC inhibitor K1 (10 μM) in cells derived from tumour tissue of DEN-treated wild-type and LDKO mice ($n = 3$ independent experiments). * Indicates significant difference, $P < 0.05$ as determined by two-tailed $t$-test. Data are represented as mean ± s.e.m.

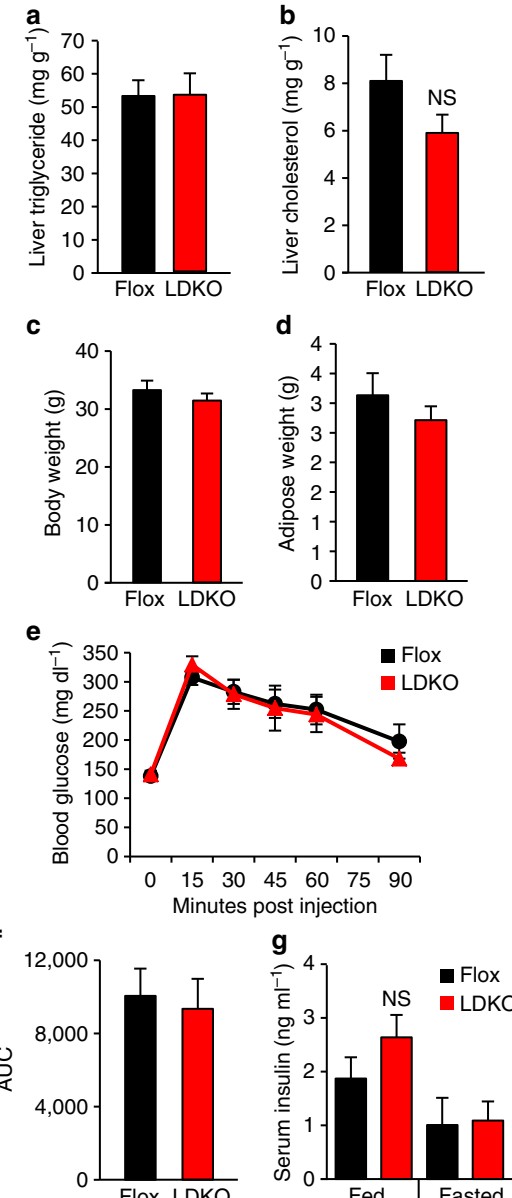

**Figure 3 | Loss of hepatic ACC activity has minimal impact on whole-body physiology in DEN-treated mice.** Liver (**a**) triglyceride and (**b**) cholesterol content, and (**c**) body weight and (**d**) adipose weight of DEN-treated LDKO and Flox mice at 40 weeks of age. (**e**) Blood glucose concentrations over time and (**f**) integrated area under the curve (AUC) during a glucose tolerance test in DEN-treated LDKO and Flox mice at 22 weeks of age. (**g**) Serum insulin levels in random-fed or 12 h-fasted Flox and LDKO mice at 22 weeks of age. For **a**–**g**, $n = 10$ Flox and 11 LDKO mice. Statistical significance was analysed by two-tailed $t$-test (**g** was analysed by one-way analysis of variance followed by Tukey's *post hoc* analysis.) Data are represented as mean ± s.e.m.

substrate for multiple antioxidant enzymes that convert oxidized glutathione (GSSG) into reduced glutathione (GSH). Accordingly, we used biochemical assays to verify that LDKO tissue had improved antioxidant status as indicated by an increase in the GSH:GSSG ratio compared to Flox control tissue (Fig. 5f). Consistent with the greater antioxidant status, metabolomics data also showed that LDKO tissues had decreased markers of oxidative stress including peroxidized lipids 13-HODE and 9-HODE, oxidized ascorbate, and oxidized methionine (Supplementary Data 1).

DEN is a pro-oxidant that induces tumorigenesis via genotoxic damage[20]; therefore, we investigated the acute effects of DEN on

DNA damage, apoptosis and proliferation in LDKO and Flox liver 24 h after exposure. Compared to Flox control liver, LDKO liver had less DEN-induced 8-hydroxydeoxyguanosine (8-OHdG) oxidative DNA adducts (Supplementary Fig. 4f) and fewer nuclei positive for γH2A.X expression (Fig. 6a,b). Furthermore, LDKO liver had less DEN-induced cleaved CASP3 expression (Fig. 6c,d) and higher Ki67 expression (Fig. 6e,f) than controls. LDKO liver tissue also maintained a higher GSH:GSSG ratio than controls and had a non-significant ($P = 0.15$, two-tailed

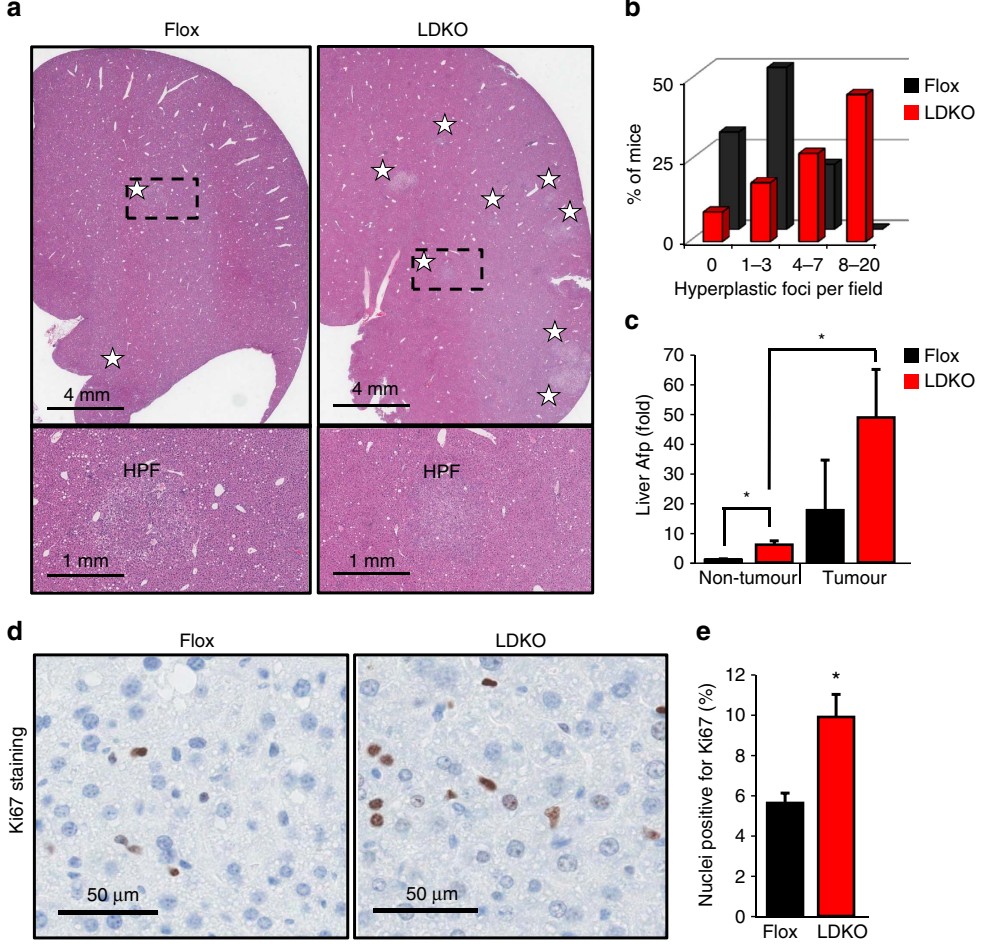

**Figure 4 | Markers of proliferation in DEN-treated mice.** (**a**) Representative haematoxylin and eosin staining of liver sections of DEN-treated Flox and LDKO mice, and (**b**) quantification of hyperplastic foci (HPF) per field (white stars indicate hyperplastic foci). For top image, scale bar, 4 mm; for bottom image, scale bar, 1 mm. (**c**) mRNA expression of alpha-fetoprotein (*Afp*) in liver non-tumour and tumour tissue of LDKO and Flox mice. (**d**) Representative images and (**e**) quantification of Ki67 immunohistochemical staining of DEN-treated Flox and LDKO mice. Data are from DEN-treated mice at 40 weeks of age. Scale bar, 50 μm. For non-tumour tissue, *n* = 10 for Flox and 11 for LDKO; for tumour tissue, *n* = 3 Flox and 6 LDKO mice. * Indicates significant difference, *P* < 0.05 as determined by two-tailed *t*-test. Data are represented as mean ± s.e.m.

Student's *t*-test) increase in NADPH levels (Supplementary Fig. 4g,h).

To assess whether the LDKO primary hepatocytes had a general antioxidant advantage compared to Flox hepatocytes, we measured cell viability in response to increasing concentrations of DEN or another pro-oxidant tert-butyl hydroperoxide (tBuOOH). Compared to Flox control cells, ACC-deficient cells had a greater ability to survive both DEN and tBuOOH, as evidenced by a rightward shift in the half-maximal inhibitory concentration (Fig. 7a,b). These data are consistent with the *in vivo* results shown in Fig. 6. To determine whether the improved antioxidant defence in LDKO hepatocytes was sufficient to protect from DEN-induced genotoxic damage and/or cell survival, primary hepatocytes from wild-type mice were treated with a cell-permeable glutathione precursor γ-glutamyl monoethyl ester (GEE) prior to DEN exposure. Importantly, GEE protected from DEN-induced DNA damage and apoptosis as evidenced by lower γH2A.X expression and decreased cleaved CASP3 (Fig. 7c,d). Ki67 data are not shown for this experiment because primary hepatocytes do not proliferate in culture. In summary, these data show that increasing antioxidant capacity of control hepatocytes protected from DEN-induced cell death and recapitulated several phenotypes of LDKO hepatocytes treated with DEN.

## Discussion

Compared to non-cancerous cells of origin, tumour cells have altered metabolic programming that facilitates aberrant proliferation and survival. Increased lipogenesis is thought to benefit tumour cells by supplying saturated lipids for membrane synthesis, activation of anabolic signalling pathways and/or protection from lipid peroxidative stress[13,27–29]. Consistent with this, blocking lipogenesis in liver cancer cells results in reduced viability[28]. However, using a mouse model that lacks hepatic lipogenesis, we show that lipogenesis is completely dispensable for liver tumorigenesis. Unexpectedly, inhibition of lipogenesis via genetic deletion of the ACC enzymes increased susceptibility to tumorigenesis and increased tumour multiplicity compared to wild-type controls. This surprising increase in tumorigenesis reveals a significant gap in knowledge concerning the role of lipogenesis in liver tumorigenesis.

The mechanism of increased liver tumorigenesis in LDKO mice was associated with a robust antioxidant defence that resulted in increased survival of DEN-damaged cells. This was evidenced by decreased apoptosis in both LDKO liver tissue and LDKO primary hepatocytes exposed to DEN. The results in primary hepatocytes showed that the antioxidant survival phenotype was cell autonomous and was not a consequence of altered metabolic milieu *in vivo*. Furthermore, the LDKO mice

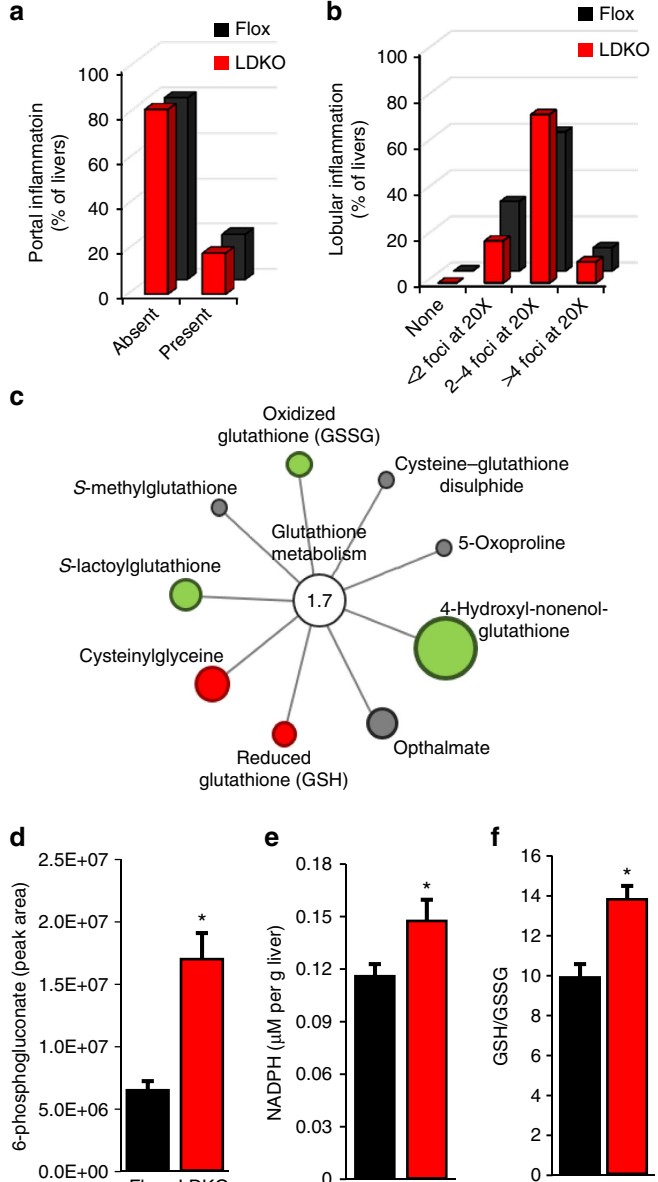

**Figure 5 | Inflammatory and metabolomic characteristics of DEN-treated liver tissues.** Histological characterization of (**a**) portal and (**b**) lobular inflammation in livers of DEN-treated LDKO and Flox mice at 40 weeks of age. (**c**) Metabolomics-based pathway enrichment analysis of glutathione metabolism in DEN-treated livers from LDKO mice relative to Flox mice at 40 weeks of age. Red colour indicates a significant increase in metabolite levels, green indicates a significant decrease in metabolite levels and grey indicates no significant difference in LDKO/Flox levels. Circle size corresponds with the relative magnitude of difference between LDKO/Flox levels. Levels of (**d**) 6-phosphogluconate, (**e**) NADPH and (**f**) GSH:GSSG in liver tissue extracts of DEN-treated LDKO and Flox mice at 40 weeks of age. * Indicates significant difference, $P < 0.05$ as determined by two-tailed $t$-test. For **a,b** and **d–f**, $n = 10$ Flox and 11 LDKO mice. For **c**, $n = 6$ mice. Data are represented as mean ± s.e.m.

did not have any genotype-specific differences in physiologic risk factors for liver tumorigenesis including liver fat content, glucose tolerance or liver inflammation that could affect tumorigenesis.

The increased antioxidant defence in LDKO tissue was evidenced by increased NADPH levels and an increased GSH/GSSG ratio. NADPH is a powerful reducing agent that is required to maintain cellular antioxidants, including glutathione, in a reduced state. Because NADPH is normally consumed by lipogenesis, it is logical that LDKO hepatocytes would have increased accumulation of NADPH. However, metabolomics data also revealed that LDKO mutant mice had increased PPP activity, which also increases NADPH production. The rate-limiting enzyme in the PPP, glucose-6-phosphate dehydrogenase, is regulated by the lipogenic transcription factor SREBP1c; therefore, it is not unexpected that PPP activity was increased when the lipogenic ACC enzymes were deleted. Importantly, the increased antioxidant defence has a functional phenotype because treatment of primary wild-type hepatocytes with the glutathione precursor GEE was sufficient to decrease DEN-induced apoptosis in wild-type hepatocytes to a similar level as LDKO hepatocytes. The antioxidant effects of ACC inhibition observed in this study are consistent with a previous study in lung cancer cells, wherein activation of AMP-activated protein kinase (an inhibitor of ACC enzymes) increased intracellular NADPH levels and the GSH:GSSG ratio[30]; however, the investigators did not measure flux through the PPP.

Although we show that improved antioxidant defence in LDKO mice contributes to the phenotype of increased liver tumorigenesis, we cannot exclude additional roles for other pathways. For example, we observed that many lipid species were altered between genotypes. One of the largest changes observed was saturation of 18-carbon fatty acids where stearate (18:0) was decreased, while the PUFAs linoleate and linolenate were increased (18:2 and 18:3, respectively). Furthermore, there was a general trend for glycerolipids including MAGs, PCs and PEs to be more polyunsaturated (18:2 and 18:3) than saturated or monounsaturated (18:0 and 18:1). This pattern may be due to less oxidative attack on the conjugated dienes of the polyunsaturated lipids, alterations in the activity of lipid desaturases, changes in fat oxidation or changes in lipid uptake by the upregulation of fatty acid transporters including CD36 and FATP5. Because LDKO hepatocytes cannot synthesize lipids and must rely on extracellular lipids, it is possible that their increased intake of lipids may facilitate tumorigenesis. We next analysed the metabolomics data for sphingolipids because they have known roles in cell survival and apoptosis. LDKO hepatocytes had normal 16:0 ceramide; however, most other sphingolipids were decreased including sphinganine and sphin-gomyelin species. The impact of these alterations in lipid composition on liver tumorigenesis is unclear and merits future investigation.

Other studies have investigated the role of lipogenesis in liver tumorigenesis using genetic models and targeting fatty acid synthase (FASN). It has been shown that liver-specific deletion of FASN completely prevents development of tumours driven by hydrodynamic-mediated overexpression of Akt either alone[31] or in combination with c-Met[32] in C57BL/6 mice. However, liver tumours driven by Akt in combination with N-Ras are resistant to FASN inhibition[33]. These data demonstrate that the role of lipogenesis in tumorigenesis may be highly dependent on the genetic drivers involved. Interestingly, a recent study showed that FASN inhibition causes breast cancer cell death independent of its effects on inhibiting lipogenesis[34]. This was evidenced by the fact that ACC inhibition could prevent cell death caused by FASN inhibition[34]. Rather, cell death may involve toxicity from malonyl-CoA accumulation. These data support the findings of the present study by showing that lipogenesis is also not necessary for breast cancer cell survival.

In summary, current evidence in the literature and our finding that lipogenesis is upregulated in DEN-induced liver tumours and cancer cell lines led us to hypothesize that inhibition of ACC enzymes would prevent the development of DEN-induced liver

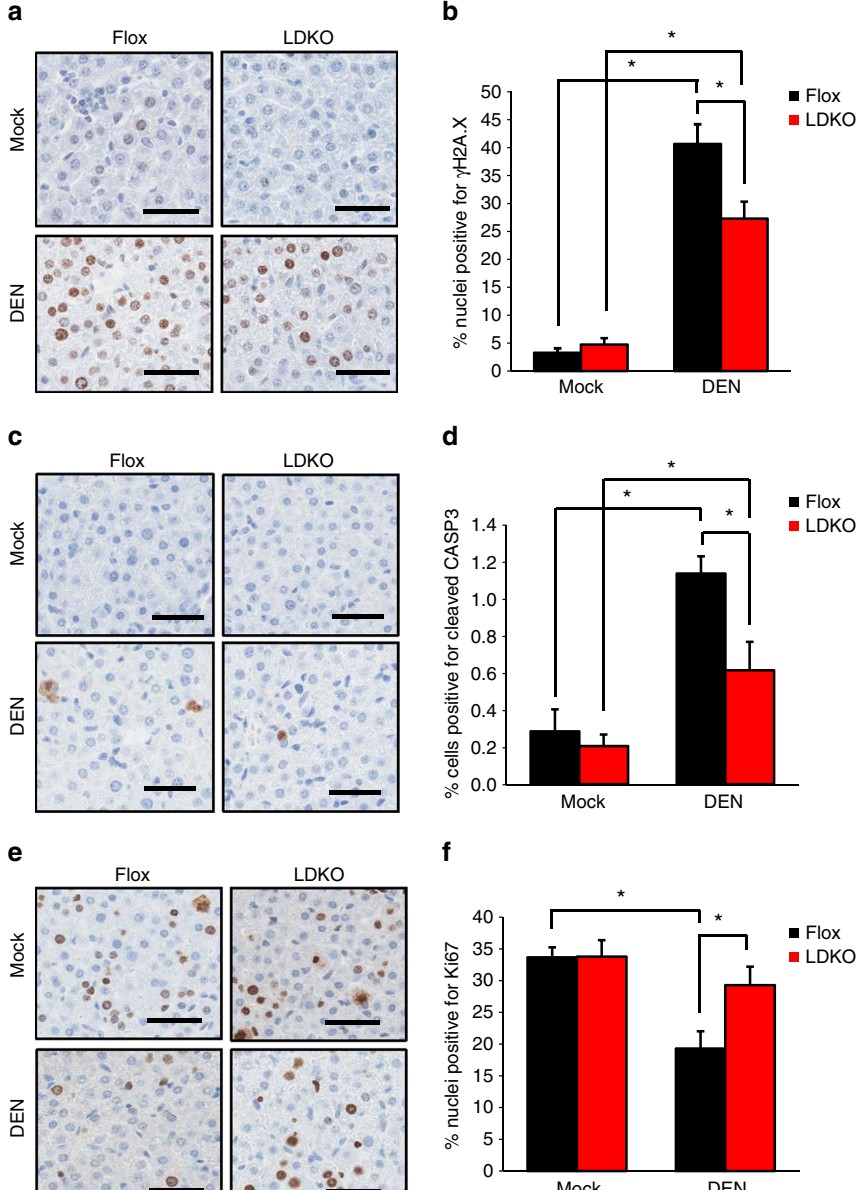

**Figure 6 | LDKO livers are resistant to DEN-induced DNA damage and apoptosis.** Representative images of immunohistochemical staining and quantification for (**a,b**) γH2A.X, (**c,d**) cleaved CASP3 and (**e,f**) Ki67 in liver sections of LDKO and Flox mice treated with vehicle (Mock) or DEN at 14 days of age and collected 24 h after treatment (scale bars, 50 μm; $n = 4$ mice). * Indicates significant difference, $P < 0.05$ as determined by one-way analysis of variance followed by Tukey's *post hoc* analyses. Data are represented as mean ± s.e.m.

tumorigenesis. However, our data clearly showed the converse. We found that lipogenesis is not only dispensable for DEN-induced tumorigenesis, but inhibition of ACC enzymes caused a surprising increase in tumorigenesis. One mechanism for this unexpected phenotype was linked to improved antioxidant defence and hepatocyte survival in LDKO cells treated with DEN. On the basis of these data, we propose that the increase in tumorigenesis in LDKO mice is a consequence of increased survival of DEN-damaged hepatocytes that later become tumour-initiating cells. Importantly, these data are timely and relevant because ACC inhibitors are in clinical development for a range of disorders including cancer[35]. Although the present study only investigated DEN-induced tumorigenesis in mice, it suggests that ACC inhibitor use in humans should be approached with caution in the context of cancer therapy where antioxidant defence and cell survival are not desirable outcomes.

## Methods

**Evaluating mRNA expression using Oncomine.** A search was conducted within Oncomine (www.oncomine.com) for 'Hepatocellular Carcinoma'. A filter for studies that reported expression of ACC1 (*ACACA*) and ACC2 (*ACACB*) was then applied. Studies that reported mRNA expression data for both HCC and non-cancerous liver tissue were identified, resulting in three independent studies[36–38], which encompassed data from a total of 327 HCC and 386 non-cancerous liver samples. One study contained clinical data for HCC grade for each sample[38], so these data were also used in a separate analysis to compare mRNA expression by tumour grade. All mRNA expression values were converted from Log$_2$ to linear scale and mRNA expression of HCC tissue was expressed as fold-difference from the non-cancerous liver within each individual study. Linear fold-expression values from the individual studies were pooled to allow overall comparison of mRNA expression in HCC compared to non-cancerous tissue.

**Cell culture.** HepG2, Hepa1-6 and Hepa-1c1c7 cells were obtained from American Type Culture Collection. Cell lines were amplified and frozen in fetal bovine serum (FBS) containing 10% dimethylsulphoxide. Thawed cells were screened for mycoplasma contamination and used within 10 passages. PH5CH8 cells were

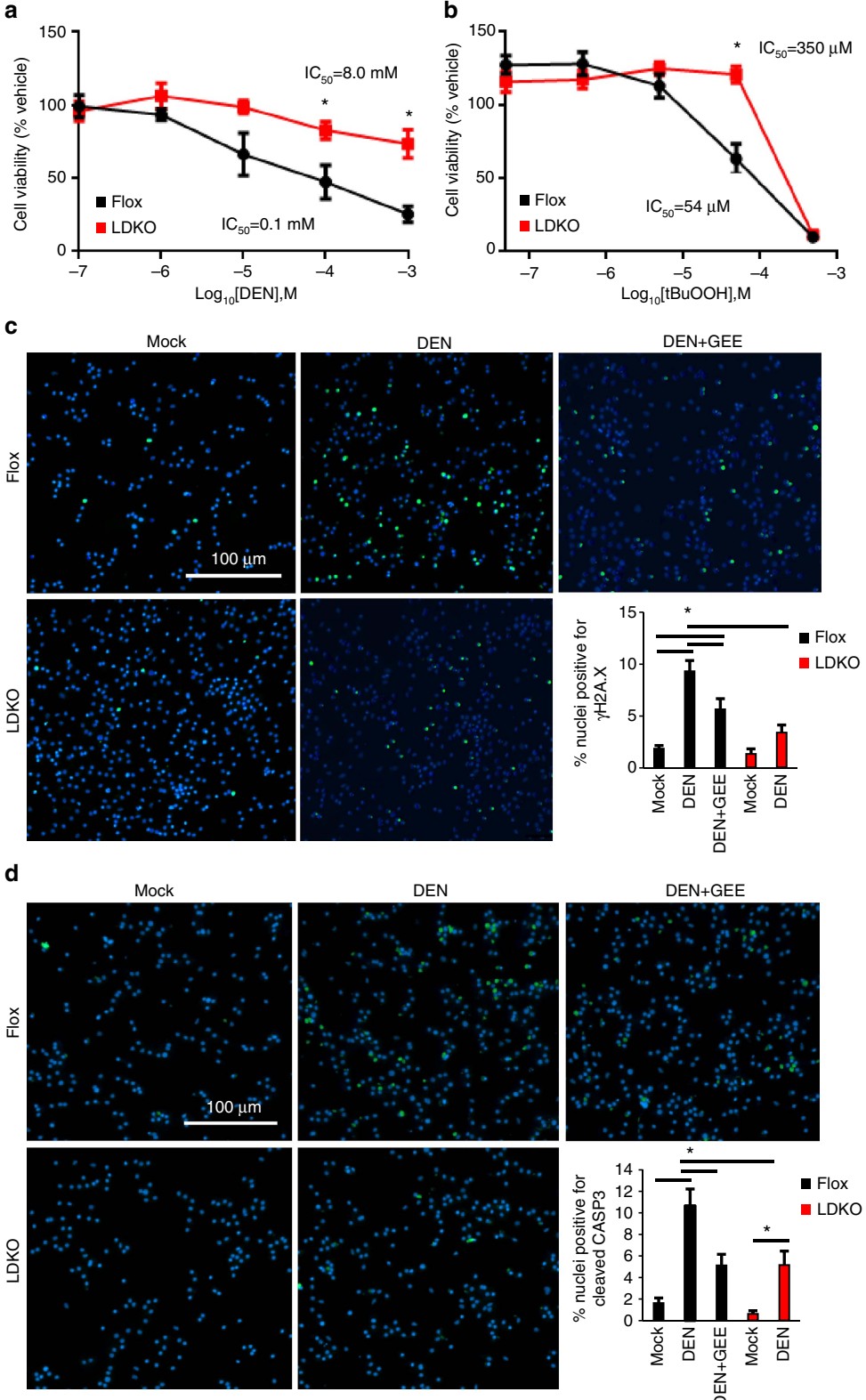

**Figure 7 | Glutathione supplementation protects hepatocytes from DEN-induced DNA damage and apoptosis.** Viability of primary hepatocytes treated with (**a**) DEN or (**b**) tBuOOH for 24 h. Representative images and quantification of immunofluorescence in primary hepatocytes treated with vehicle (Mock) or DEN for 24 h, or pretreated with GEE (1 mM) for 1 h prior to DEN treatment, probed with primary antibodies against (**c**) γH2A.X and (**d**) cleaved CASP3. Nuclei were counter-stained with 4,6-diamidino-2-phenylindole. Scale bar, 100 μm. Quantification for **a–d** includes three and two independent experiments, respectively. * Indicates significant difference, $P < 0.05$ as determined by one-way analysis of variance followed by Tukey's *post hoc* analyses. Data are represented as mean ± s.e.m.

provided by Dr Young Hahn, University of Virginia. Huh7 cells were provided by Dr Geoffrey Girnun, Stony Brook School of Medicine, New York. Primary mouse hepatocytes were isolated as described[17], seeded at $1.5 \times 10^5$ cells per ml in DMEM supplemented with $4.5 \, g \, l^{-1}$ glucose, 10% FBS, $1 \, \mu M$ dexamethasone and $100 \, nM$ insulin for 4 h. For experiments testing the effects of DEN in vitro, primary hepatocytes were isolated from 2-week-old Flox and LDKO mice and seeded at $3 \times 10^5$ cells per ml onto collagen-coated glass coverslips. Four hours after isolation, the media was changed to DMEM supplemented with $1 \, g \, l^{-1}$ glucose, 0.2% bovine serum albumin, $100 \, nM$ dexamethasone and $1 \, nM$ insulin. For viability experiments, cells were treated with DEN or tBuOOH for 24 h, then an MTT (3-(4,5-dimethylthiazol-2-yl)-2,5-diphenyltetrazolium bromide) assay was performed, and absorbance at 570 nm was measured. For rescue experiments, cells were treated with vehicle or 1 mM glutathione monoethyl ester (GEE; cell-permeable glutathione precursor, Bachem H-1298) for 1 h, then vehicle or $50 \, \mu M$ DEN was added. Cells were fixed 24 h after DEN with 4% paraformaldehyde in PBS (pH 7.4) at room temperature for 10 min. Cells were permeabilized with 0.25% Triton X-100 in PBS. Cells were treated with boiling-hot citrate antigen buffer (10 mM Na-citrate, pH 6.0), blocked (1% basal serum albumin (BSA), $22.25 \, mg \, ml^{-1}$ glycine, 0.1% Tween 20), incubated in primary antibody at a concentration of 1:500 in 1% BSA overnight at $4 \, ^{\circ}C$, then incubated with green fluorescent protein-conjugated secondary antibody at a concentration of 1:5,000. Nuclei were counter-stained using $0.1 \, \mu g \, ml^{-1}$ 4,6-diamidino-2-phenylindole. Primary antibodies used were: Ki67 (Santa Cruz 7846), cleaved CASP3 (Cell Signaling 9661) and $\gamma$H2A.X (Cell Signaling 9718S).

**In vitro lipogenesis assays.** Lipogenesis assays were performed according to methods previously established in our laboratory[39]. In brief, assays were performed on monolayered cells in Krebs-Ringer Phosphate buffer supplemented with non-labelled substrates: 5 mM glucose, 0.5 mM glutamine, $125 \, \mu M$ palmitate, $50 \, \mu M$ acetate and 1 mM carnitine with addition of either $5 \, \mu Ci \, ml^{-1}$ $^{14}$C-glucose or $10 \, \mu Ci \, ml^{-1}$ $^{14}$C-acetate.

**Seahorse extracellular flux assays.** Measurement of oxygen consumption rate (OCR) and extracellular acidification rate of cells during a mitochondrial stress test was performed using a Seahorse XF-24 Flux Analyzer (Seahorse Biosciences), as we have previously established in our laboratory[39]. In brief, cells were seeded in DMEM supplemented with $4.5 \, g \, l^{-1}$ glucose and 10% FBS 24 h before assay. One hour before assay the media was changed to sodium bicarbonate-free DMEM and the cells were placed in a $37 \, ^{\circ}C$ incubator without $CO_2$ buffering. Cells were treated with oligomycin, BAM15, antimycin A or rotenone at the times and concentrations indicated. Mitochondrial spare respiratory capacity represents the difference between the maximal OCR upon addition of the mitochondrial uncoupler BAM15 (ref. 40) and the basal OCR at 18 min before addition of oligomycin.

**Mice.** All mice were housed and bred in University of Virginia vivarium facilities. Mice were maintained in a temperature-controlled room ($22 \, ^{\circ}C$) on a 12-h light/dark cycle in filter-top cages and with ad libitum access to food and water. All animal studies were approved by the University of Virginia Institutional Animal Care and Use Committee and performed according to criteria outlined in the 'Guide for the Care and Use of Laboratory Animals' prepared by the National Academy of Sciences and published by the National Institutes of Health (NIH publication 86-23 revised 1985). To assess ACC expression in livers and tumours of mice treated with DEN, male C57BL/6N mice were treated with $25 \, mg \, kg^{-1}$ DEN at 14 days of age via intraperitoneal injection as we have previously established in our laboratory[23]. Mice were weaned at 21 days of age fed a normal chow diet prepared in-house according to methods previously established by our laboratory[23] until final tissue collection at 32 weeks of age. To study the effect of liver ACC1 and ACC2 genetic deletion in mice, LDKO mice were generated on a C57BL/6J background, as we have previously described[17] and compared with floxed (Flox) littermate controls. To determine whether loss of ACC expression could decrease or increase tumour incidence, we used female mice for this study because tumour incidence is ~30%, whereas for males tumour incidence is ~90%. Sample size sufficient to detect a 20% change in tumour number was estimated a priori, using a power analysis based on group means and s.d.'s previously reported[22,23]. Offspring female LDKO and Flox mice were treated with $25 \, mg \, kg^{-1}$ DEN at 14 days of age via intraperitoneal injection. For the long-term DEN study, mice were genotyped and paired with littermates and weaned to cages at 21 days of age, $n = 10$ Flox and 11 LDKO mice. Starting at 6 weeks of age, mice were fed a western diet (D12451, Research Diets), which contained (w/w) 21% lard and 20% sucrose. Tumour burden analysis and final tissue collection were performed at 40 weeks of age. For the acute DEN study, mice were genotyped at 12 days of age, treated with $25 \, mg \, kg^{-1}$ DEN at 14 days of age and then tissues were collected 24 h after DEN treatment for analysis. Experimenters were blinded to genotype and/or treatment during assessment of experimental outcomes.

**Tissue harvest and tumour analysis.** Mice were killed in the random-fed state between 9:00 and 11:00 h. For tumour multiplicity, the number of surface-hemorrhaging tumours per liver was counted. The large lobe of each liver was fixed in 10% neutral-buffered formalin for paraffin-embedding. The remaining liver was

divided into non-tumour-involved and tumour-involved tissue, snap-frozen in liquid nitrogen and stored at $-80 \, ^{\circ}C$ until further biochemical analysis. For adiposity measurements, the weights of both subcutaneous and gonadal fat pads were summed for each animal.

**Histology and immunohistochemistry.** The large lobe of the liver was fixed in 10% formalin and paraffin-embedded for microtome sectioning ($5 \, \mu m$) and haematoxylin and eosin staining or immunohistochemistry (IHC). Slides were digitally scanned using an Aperio ScanScope (SC System) to produce high-resolution images (resolution: $0.25 \, \mu m$ per pixel). Histological analysis and IHC quantitation was performed in a blinded manner. Hyperplastic foci were histologically characterized by the presence of mitotic figures and hyperbasophilia. Tumours were histologically characterized according to methods previously established[23]. Hepatocellular adenoma (HCA) was characterized by the presence of basophilic cells and cells containing glycogen and fat, resembling human HCA. HCC was distinguished from pre-neoplastic lesions if three or more of the standard criteria were met: undifferentiated trabecular structure; enlarged, mild-moderately polymorphic hyperchromatic nuclei with enlarged nucleoli; presence of basophilia; increased abundance of mitotic figures; and invasive growth. IHC quantitation was performed on six randomly selected $3 \, cm^2$ areas for each animal.

**Liver fat content.** Lipids were extracted from 50 mg non-tumour-involved liver tissue using 2:1 chloroform:methanol as previously described[41]. Lipid extract was dried down under nitrogen gas and resuspended in 0.4 ml of 95% ethanol. Resuspended lipid was used for colorimetric assays to measure triglyceride (Pointe Scientific) at a 1:5 in-assay dilution and cholesterol (Infinity, Thermo Scientific) at a 1:2 assay dilution according to the manufacturers' protocols.

**Metabolomics.** Metabolomics analyses were performed by Metabolon, Inc. (Durham, North Carolina, USA) as previously described[42,43] from 30 mg frozen liver tissue. Samples were stored at $-80 \, ^{\circ}C$ until processed. In brief, protein was precipitated with methanol under vigorous shaking for 2 min. Recovery standards were added at this step for quality control purposes. The supernatant was divided into four fractions: one for analysis by ultra-high-performance liquid chromatography–tandem mass spectrometry (UPLC–MS/MS; positive ionization), one for analysis by UPLC–MS/MS (negative ionization), one for the UPLC–MS/MS polar platform (negative ionization) and one for analysis by gas chromatography–mass spectrometry. Instrument variability was determined to be 5% by calculating the median relative standard deviation for the standards that were added to each sample before injection into the mass spectrometers. Overall, process variability was determined to be 8% by calculating the median relative standard deviation for all endogenous metabolites (that is, non-instrument standards) present in 100% of the pooled mouse liver samples. Experimental samples and controls were randomized across the platform run.

**Western blotting.** Frozen liver tissue was homogenized in RIPA buffer (50 mM Tris, 100 mM NaCl, pH 8.0, 10 mM EDTA, pH 7.0, 0.4% v/v Triton X-100, 10 mM nicotinamide), containing protease inhibitor cocktail (Roche) and phosphatase inhibitors (2 mM sodium orthovanadate, 1 mM sodium pyrophosphate, 10 mM sodium fluoride, 250 nM microcystin LR). Cells were lysed in HES-SDS lysis buffer (250 mM sucrose, 20 mM HEPES, pH 7.4, 1 mM EDTA, 2% SDS). Lysates were centrifuged at $16,000 \, g$ at $4 \, ^{\circ}C$ for 10 min. Laemmli buffer was added to supernatant, and samples were heated to $65 \, ^{\circ}C$ for 10 min. Proteins were resolved using SDS–polyacrylamide gel electrophoresis and transferred onto nitrocellulose membrane. Membranes were blocked in 5% milk, incubated with primary antibody at a concentration of 1:1,000 in TBST with 5% BSA overnight at $4 \, ^{\circ}C$, then incubated with fluorescent-tagged secondary antibody at a concentration of 1:10,000 and read using a LI-COR Odyssey infrared imaging system. Primary antibodies used were: pan-ACC (Cell Signaling 3676), Ki67 (Epitomics 4203 for IHC), Cleaved CASP3 (Cell Signaling 9661), $\gamma$H2A.X (Cell Signaling 9718S) and 14-3-3 (Santa Cruz 1657).

**Real-time quantitative PCR.** RNA was extracted from frozen liver tissue using ice-cold TRIzol reagent (Life Technologies) according to the manufacturer's protocol, followed by DNase I treatment (Roche). Complementary DNA was synthesized by standard reverse transcription (High Capacity cDNA synthesis kit, Roche) on a MultiGene thermal cycler (Labnet). Amplification and semi-quantification of transcripts were performed using Sensifast SYBR Green mix (Bioline) on a real-time PCR system (iCycler, Bio-Rad) with mouse gene-specific primers (Integrated DNA Technologies). Cyclophilin A (CypA) was used as a housekeeping gene. Primers used were CypA (F: 5′-CGATGACGAGCCCTTGG-3′, R: 5′-TCTGCTGTCTTTGGAACTTTGTC-3′), alpha-fetoprotein (Afp; F: 5′-CCC GCTTCCCTCATCC-3′, R: 5′-GAAGCTATCCCAAACTCATTTTCG-3′), glucose-6-phosphate dehydrogenase (G6pd; F: 5′-AAGAAGCCTGGCATGTTCTT-3′, R: 5′-GAAGCCCACTCTCTTCATCA-3′), lipoprotein lipase (Lpl; F: 5′-GGATGG ACGGTAACGGGAAT-3′, R: 5′-ATAATGTTGCTGGGCCCGAT-3′), Cd36 (F: 5′-GATGACGTGGCAAAGAACAG-3′, R: 5′-TCCTCGGGGTCCTGAGTTAT-3′),

fatty acid transport protein 5 (*Fatp5*; F: 5′-GCACCTTCTGACCCAGTACC-3′, R: 5′-GTAAGCAGCCAAGGAATCCA-3′).

**NADPH assays.** An amount of 20 mg of frozen liver was homogenized and 75 μg was used per well for NADPH measurement using an enzyme cycling assay kit (Abcam 65349) according to the manufacturer's protocol.

**Glutathione assay.** GSH and GSSG were extracted from 40 mg of frozen liver tissue and levels were measured using a fluorescent detection kit (BioVision K264) according to the manufacturer's protocol. Before the assay, spin columns (EMD Millipore, 10 kDa) were used to eliminate interfering proteins.

**8-OHdG assay.** Genomic DNA was extracted from 50 mg frozen liver tissue using a purification spin column followed by DNA elution according to manufacturer's instructions (Qiagen AllPrep DNA/RNA/Protein Mini). Extracted DNA was converted to single-stranded DNA by incubation at 95 °C for 5 min then rapidly chilled on ice. Single-stranded DNA was then digested to nucleosides in 20 mM sodium acetate buffer pH 5.2 containing nuclease P1 (Sigma N8630) for 2 h, followed by incubation in 100 mM Tris buffer containing alkaline phosphatase (Sigma P6774) at a final pH of 7.5 at 37 °C for 1 h. The nucleoside-containing mixture was centrifuged at 6,000 g for 5 min and the supernatant was used for the 8-OHdG assay. 8-OHdG was quantitated using an enzyme-linked immunosorbent assay according to the manufacturer's protocol (Cell Biolabs STA-320).

**Statistical analyses.** Results are presented as mean ± s.e.m. and compared by two-tailed parametric Student's *t*-test unless otherwise indicated. Statistical significance was accepted at $P < 0.05$. Statistical analyses were performed using GraphPad Prism version 6.00 for Windows (GraphPad Software).

**Data availability.** All data supporting the findings of this study are available within the article and its Supplementary Information files or from the corresponding author upon reasonable request. The Oncomine data referenced during the study are available in a public repository from the Oncomine website (www.oncomine.com).

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

## Acknowledgements

Dr Ryan D. Michalek (Metabolon, Inc., Durham, NC, USA) provided analysis of the metabolomics data. All IHC was performed by the University of Virginia Biorepository and Tissue Research Facility under the expert direction of Dr Pat Pramoonjago. This work was funded in part by awards to K.L.H. from the US National Institutes of Health and National Health and Medical Research Council of Australia.

## Author contributions

M.E.N., J.D.Y.C., J.K.S.-D., S.H.C. and K.L.H. designed the study. M.E.N., J.D.Y.C., F.L.B., S.L., S.R.H., D.S.B., E.M.O. and K.L.H. performed experiments. L.E.W., D.E.J., G.J.C. and N.T. assisted with methodology and contributed resources. C.L. scored histology. M.E.N. and K.L.H. wrote and edited the manuscript with input from the other authors.

## Additional information

**Competing financial interests:** The authors declare no competing financial interests.

