## [Peer Review File · Nature Communications]

Reviewers' comments:

Reviewer #1 (Remarks to the Author):

Healy et al have examined the effect of ACC1/2 deletion on liver tumorigenesis in mice and surprisingly determined that inhibition of hepatic lipogenesis via this route does not prevent, but rather enhances tumor growth. This is surprising given previous indications that increased de novo lipogenesis was an important adaptation in tumor cells to generate biomass for new membranes. The studies are well done and the story is told well. I have a few moderate concerns.

The experiment to examine tumor formation in vivo seems to have been done with 10 and 11 mice per group. That doesn't seem like a lot of mice per group. Is this sufficiently powered to detect a difference that is statistically meaningful? It would also provide more confidence if a cell proliferation assay were done in vitro.

The observation that the knockouts are resistant to DEN-induced cell death likely due to enhanced anti-oxidant capacity is interesting. Are the cells resistant to other forms of oxidative damage and induced cell death? This would increase the confidence in the mechanism.

Reviewer #2 (Remarks to the Author):

In the present study, Healy et al. inhibited hepatic lipogenesis in mice by liver-specific knockout of acetyl-CoA carboxylase (ACC) genes and evaluated hepatocarcinogenesis potential following DEN administration. Unexpectedly, mice lacking hepatic lipogenesis exhibited increase in tumor incidence and multiplicity compared to controls. In particular, ACC-deficient livers showed an increase in antioxidant compounds including NADPH and reduced glutathione. The authors conclude that lipogenesis is dispensable for liver tumorigenesis and that ACC enzymes play a crucial role in redox regulation and cell survival in the liver.

The study by Healy et al. is novel, well-performed, and intriguing. The data are solid and support the conclusions drawn. Figures are easy to understand as well. The implications of the present study are high both in terms of the molecular pathogenesis of HCC and experimental therapeutics against this deadly disease.

Minor issues:

1. The authors conclude that lipogenesis is dispensable for liver tumorigenesis. The latter conclusions of the present study might be too simplistic and one-sided, as they completely rely on the DEN model that, although similar to human HCC with poor outcome at the molecular level, does not necessarily recapitulate the whole spectrum of human HCC. In other words, the authors should consider that lipogenesis might be either necessary or dispensable for liver carcinogenesis depending on the context or the molecular mechanisms involved. In support of this hypothesis, it has been recently shown that genetic inactivation of fatty acid synthase completely suppresses hepatocarcinogenesis induced by overexpression of AKT, either alone or in association with c-Met, in mice (Li et al., *J Hepatol.* 2016;64:333-41; Hu et al., *Sci Rep.* 2016 Feb 9;6:20484). On the other hand, mice overexpressing activated forms of AKT and N-Ras are completely resistant to fatty acid synthase depletion in terms of hepatocarcinogenesis (Li et al. *Hepatology* 2016;63:1900-13). The authors should cite and discuss extensively these models in the Discussion section of the manuscript.

2. Another route of lipogenesis in HCC has been shown to be dependent on lipoprotein-lipase (Cao et al., *Liver Int.* 2016 Jun 6). The authors should demonstrate in their models whether LPL is

induced following ACC knockdown.

Reviewer #3 (Remarks to the Author):

The authors present an interesting study which reports that liver lipogenesis does not contribute to HCC. Instead, HCC is characterized by protective redox regulation response.

The metabolomics methodology is based on well-established commercial platform yet it is described too briefly. More detail would be needed, particularly as there is much difference when preparing liver tissue or serum samples for the analysis. The method reference provided is for serum sample analyses. How were the liver tissue samples prepared, how much sample was used etc etc.?

The overall study setting is adequate given the study aims. Unexpected results are reported, making this study all the more interesting and potentially important.

Lipogenesis is a hallmark of NAFLD, and at the epidemiological level, there is clear association between increased prevalence of NAFLD and HCC. Nevertheless, obesity-related NAFLD ('metabolic NAFLD') is associated with specific lipid profile, unlike e.g. in PNPLA3-associated NAFLD. This specific lipid profile is characterized by increased ceramides (which have a role in apoptosis), diacylglycerols, and triacylglycerols with low carbon number and double bond content.

In order to truly understand the role of liver lipogenesis in HCC development, it would therefore be also important to understand if ACC inhibition in the present study affects the metabolism of the key reactive lipids associated with 'metabolic NAFLD'. I would therefore suggest that the authors examine liver tissue in their studies by also performing lipidomic analyses.

Response to Reviewers

Manuscript: Healy *et al.*, NCOMMS-16-16802

Reviewer 1:

Healy et al have examined the effect of ACC1/2 deletion on liver tumorigenesis in mice and surprisingly determined that inhibition of hepatic lipogenesis via this route does not prevent, but rather enhances tumor growth. This is surprising given previous indications that increased de novo lipogenesis was an important adaptation in tumor cells to generate biomass for new membranes. The studies are well done and the story is told well. I have a few moderate concerns.

The experiment to examine tumor formation *in vivo* seems to have been done with 10 and 11 mice per group. That doesn't seem like a lot of mice per group. Is this sufficiently powered to detect a difference that is statistically meaningful? It would also provide more confidence if a cell proliferation assay were done *in vitro*.

The observation that the knockouts are resistant to DEN-induced cell death likely due to enhanced anti-oxidant capacity is interesting. Are the cells resistant to other forms of oxidative damage and induced cell death? This would increase the confidence in the mechanism.

Response 1: Regarding animal numbers. Thank you for the question. We compared our study n's to other studies using the DEN model. We find that our n of 10-11 is consistent with other studies published in quality journals (see below). Importantly, our data identify statistically significant differences and no outliers were removed from any of our data.

1. Michael Karin, *Cell* 2010, n=10-12, primary outcomes: multiplicity, size and incidence. Doi: 10.1016/j.cell.2009.12.052.¹
2. Michael Karin, *Cell* 2005, n=9-10, primary outcomes: incidence, multiplicity, size, and proliferation. Doi: 10.1016/j.cell.2005.04.014.²
3. Naoko Ohtani, *Nature* 2013, n=6-19, primary outcomes: multiplicity and size. doi: 10.1038/nature12347.³
4. Wafik El-Deiry, *JCI* 2007, n=10, primary outcomes: tumor burden, proliferation, apoptosis doi:10.1172/JCI29900.⁴
5. Robert F Schwabe, *Gut* 2011, n=8-12, primary outcomes: multiplicity, size. doi:10.1136/gut.2010.209551.⁵
6. Michael Karin, *PNAS* 2006, n=6-10, primary outcomes: multiplicity, size, proliferation, apoptosis. doi: 10.1073/pnas.0603499103.⁶

Response 2: It would also provide more confidence if a cell proliferation assay were done *in vitro*.

Hepatocytes can be highly proliferative *in vivo* and regenerate nearly normal liver mass within 5-7 days after 2/3 partial hepatectomy (Reviewed⁷ Doi: 10.2353/ajpath.2010.090675). However, primary hepatocytes in culture do not proliferate and undergo rapid dedifferentiation. Therefore, we have not been able to measure proliferation in culture. We have revised the manuscript based on this reviewers comment. Because primary hepatocytes do not proliferate, we have now removed former Fig 7C (Ki67 in primary hepatocytes that showed no statistically significant genotype-specific difference in staining) from the manuscript to avoid confusion that primary LDKO hepatocytes proliferate in culture.

Response 3: Are the cells resistant to other forms of oxidative damage and induced cell death? This would increase the confidence in the mechanism.

This is a good question. We have now isolated primary Flox and LDKO hepatocytes and performed cell viability assays with 2 forms of oxidative damage, including tert-butyl hydroperoxide (tBuOOH) and DEN. Compared to Flox hepatocytes, LDKO hepatocytes were more resistant to cell death induced by both of these pro-oxidants. These data have been added to the revised manuscript as Figures 7A-B. These results are consistent with the overall finding that lack of ACC activity results in increased antioxidant defense and cell survival.

Reviewer 2:

In the present study, Healy et al. inhibited hepatic lipogenesis in mice by liver-specific knockout of acetyl-CoA carboxylase (ACC) genes and evaluated hepatocarcinogenesis potential following DEN administration. Unexpectedly, mice lacking hepatic lipogenesis exhibited increase in tumor incidence and multiplicity compared to controls. In particular, ACC-deficient livers showed an increase in antioxidant compounds including NADPH and reduced glutathione. The authors conclude that lipogenesis is dispensable for liver tumorigenesis and that ACC enzymes play a crucial role in redox regulation and cell survival in the liver.

The study by Healy et al. is novel, well-performed, and intriguing. The data are solid and support the conclusions drawn. Figures are easy to understand as well. The implications of the present study are high both in terms of the molecular pathogenesis of HCC and experimental therapeutics against this deadly disease.

Minor issues:

1. The authors conclude that lipogenesis is dispensable for liver tumorigenesis. The latter conclusions of the present study might be too simplistic and one-sided, as they completely rely on the DEN model that, although similar to human HCC with poor outcome at the molecular level, does not necessarily recapitulate the whole spectrum of human HCC. In other words, the authors should consider that lipogenesis might be either necessary or dispensable for liver carcinogenesis depending on the context or the molecular mechanisms involved. In support of this hypothesis, it has been recently shown that genetic inactivation of fatty acid synthase completely suppresses hepatocarcinogenesis induced by overexpression of AKT, either alone or in association with c-Met, in mice (Li et al., J Hepatol. 2016;64:333-41; Hu et al., Sci Rep. 2016 Feb 9;6:20484). On the other hand, mice overexpressing activated forms of AKT and N-Ras are completely resistant to fatty acid synthase depletion in terms of hepatocarcinogenesis (Li et al. Hepatology 2016;63:1900-13). The authors should cite and discuss extensively these models in the Discussion section of the manuscript.
2. Another route of lipogenesis in HCC has been shown to be dependent on lipoprotein-lipase (Cao et al., Liver Int. 2016 Jun 6). The authors should demonstrate in their models whether LPL is induced following ACC knockdown.

Response to point 1: The reviewer raises a good point. To be more specific about the model system used, we have revised the last sentence of the abstract to say, "This study shows that lipogenesis is dispensable for liver tumorigenesis in mice treated with DEN, and identifies an important role for ACC enzymes in redox regulation and cell survival." We have also expanded the Discussion section to address the recent studies in this area, as suggested by the Reviewer. The revised Discussion starts on page 12 and is highlighted in blue.

Response to point 2: As requested, we measured LPL expression by qPCR and Western blot. Our data showed that LPL protein and mRNA expression were not different between Flox and LDKO liver. However, we agree with the reviewer that the ACC deficient hepatocytes are likely to scavenge more lipids from the circulation so we also measured mRNA expression of the fatty acid transporters CD36 and FATP5. Both of these genes were found to be significantly increased in the LDKO livers. These data have been added as Figs. S4A-E. The Results and Discussion sections have been revised accordingly.

Reviewer 3:

The authors present an interesting study which reports that liver lipogenesis does not contribute to HCC. Instead, HCC is characterized by protective redox regulation response.

The metabolomics methodology is based on well-established commercial platform yet it is described too briefly. More detail would be needed, particularly as there is much difference when preparing liver tissue or serum samples for the analysis. The method reference provided is for serum sample analyses. How were the liver tissue samples prepared, how much sample was used etc etc.?

The overall study setting is adequate given the study aims. Unexpected results are reported, making this

study all the more interesting and potentially important.

Lipogenesis is a hallmark of NAFLD, and at the epidemiological level, there is clear association between increased prevalence of NAFLD and HCC. Nevertheless, obesity-related NAFLD ('metabolic NAFLD') is associated with specific lipid profile, unlike e.g. in PNPLA3-associated NAFLD. This specific lipid profile is characterized by increased ceramides (which have a role in apoptosis), diacylglycerols, and triacylglycerols with low carbon number and double bond content.

In order to truly understand the role of liver lipogenesis in HCC development, it would therefore be also important to understand if ACC inhibition in the present study affects the metabolism of the key reactive lipids associated with 'metabolic NAFLD'. I would therefore suggest that the authors examine liver tissue in their studies by also performing lipidomic analyses.

Response 1: Regarding metabolomics. The metabolomics method has been updated with the following text, "Metabolomics analyses were performed by Metabolon, Inc. (Durham, North Carolina, USA) as previously described^{8,9} from 30 mg frozen liver tissue. Samples were stored at -80°C until processed. Briefly, protein was precipitated with methanol under vigorous shaking for 2 min. Recovery standards were added at this step for quality control purposes. The supernatant was divided into four fractions: one for analysis by ultra-high performance liquid chromatography-tandem mass spectrometry (UPLC-MS/MS; positive ionization), one for analysis by UPLC-MS/MS (negative ionization), one for the UPLC-MS/MS polar platform (negative ionization) and one for analysis by gas chromatography-mass spectrometry (GC-MS). Instrument variability was determined by calculating the median relative standard deviation (RSD) for the standards that were added to each sample prior to injection into the mass spectrometers (median RSD typically = 4–6%; n ≥ 30 standards). Overall process variability was determined by calculating the median RSD for all endogenous metabolites (i.e., non-instrument standards) present in 100% of the pooled human plasma or client matrix samples (median RSD = 8–12%; n = several hundred metabolites)."

Response 2: Regarding request for lipidomics. Our metabolomics dataset contained dozens of lipid or lipid-related metabolites including fatty acids, phospholipids, glycerolipids and sphingolipids. We have now extracted these data to create new Figure S3A-G. These data revealed distinct patterns in lipid levels based on fatty acid chain lengths and saturations. For example, 18:0 and 18:1 fatty acids, glycerolipids and sphingolipids are typically decreased in LDKO livers whereas 18:2 and 18:3 lipids are typically increased. Additionally, sphingolipids and metabolites involved in sphingolipid metabolism tended to be lower in LDKO livers, but 16:0 ceramide was unchanged. Although we would have liked to survey the entire lipidome in more detail, the quote for this service by Metabolon was \$14,000; which is beyond our capability. Unfortunately, my laboratory is no longer capable of lipidomic analysis because of our recent move overseas and loss of key personnel. Although lipidomics could be informative, the outcomes of having lipidomics would not influence our conclusion that lipogenesis is not required for DEN-induced liver tumorigenesis. We have revised the Discussion to address the Reviewers important point about possible roles for lipid saturation and bioactive lipid species including ceramides in cell survival (pages 11-12 yellow highlight).

References:

- 1 Park, E. J. *et al.* Dietary and genetic obesity promote liver inflammation and tumorigenesis by enhancing IL-6 and TNF expression. *Cell* **140**, 197-208, doi:10.1016/j.cell.2009.12.052 (2010).
- 2 Maeda, S., Kamata, H., Luo, J. L., Leffert, H. & Karin, M. IKKbeta couples hepatocyte death to cytokine-driven compensatory proliferation that promotes chemical hepatocarcinogenesis. *Cell* **121**, 977-990, doi:10.1016/j.cell.2005.04.014 (2005).
- 3 Yoshimoto, S. *et al.* Obesity-induced gut microbial metabolite promotes liver cancer through senescence secretome. *Nature* **499**, 97-101, doi:10.1038/nature12347 (2013).
- 4 Finnberg, N., Klein-Szanto, A. J. & El-Deiry, W. S. TRAIL-R deficiency in mice promotes susceptibility to chronic inflammation and tumorigenesis. *The Journal of clinical investigation* **118**, 111-123, doi:10.1172/JCI29900 (2008).

- 5 Kluwe, J. *et al.* Absence of hepatic stellate cell retinoid lipid droplets does not enhance hepatic fibrosis but decreases hepatic carcinogenesis. *Gut* **60**, 1260-1268, doi:10.1136/gut.2010.209551 (2011).
- 6 Sakurai, T., Maeda, S., Chang, L. & Karin, M. Loss of hepatic NF-kappa B activity enhances chemical hepatocarcinogenesis through sustained c-Jun N-terminal kinase 1 activation. *Proceedings of the National Academy of Sciences of the United States of America* **103**, 10544-10551, doi:10.1073/pnas.0603499103 (2006).
- 7 Michalopoulos, G. K. Liver regeneration after partial hepatectomy: critical analysis of mechanistic dilemmas. *Am J Pathol* **176**, 2-13, doi:10.2353/ajpath.2010.090675 (2010).
- 8 Modesitt, S. C. *et al.* Women at extreme risk for obesity-related carcinogenesis: Baseline endometrial pathology and impact of bariatric surgery on weight, metabolic profiles and quality of life. *Gynecologic oncology* **138**, 238-245, doi:10.1016/j.ygyno.2015.05.015 (2015).
- 9 Evans, A. M., DeHaven, C. D., Barrett, T., Mitchell, M. & Milgram, E. Integrated, nontargeted ultrahigh performance liquid chromatography/electrospray ionization tandem mass spectrometry platform for the identification and relative quantification of the small-molecule complement of biological systems. *Anal Chem* **81**, 6656-6667, doi:10.1021/ac901536h (2009).

REVIEWERS' COMMENTS:

Reviewer #1 (Remarks to the Author):

My concerns were addressed.

Reviewer #2 (Remarks to the Author):

The authors have satisfactorily addressed the issues raised by the reviewer.

Minor issue: When speaking about the AKT and AKT/c-Met models in the Discussion section of the manuscript, the authors speak about "viral-mediated overexpression...". The authors should replace the previous sentence with "hydrodynamic-mediated overexpression" since overexpression of these genes in mice was achieved via hydrodynamic gene delivery of naked DNA, without use of viruses.

Reviewer #3 (Remarks to the Author):

The authors have adequately addressed my comments concerning the analytical method description and lipids, and I have no further comments.

Response to Reviewers

Manuscript: Healy *et al.*, NCOMMS-16-16802

Reviewer 1: My concerns were addressed.

Reviewer 2: The authors have satisfactorily addressed the issues raised by the reviewer.
Minor issue: When speaking about the AKT and AKT/c-Met models in the Discussion section of the manuscript, the authors speak about "viral-mediated overexpression...". The authors should replace the previous sentence with "hydrodynamic-mediated overexpression" since overexpression of these genes in mice was achieved via hydrodynamic gene delivery of naked DNA, without use of viruses.

Author response: As requested, we have replaced the word viral with hydrodynamic.

Reviewer 3: The authors have adequately addressed my comments concerning the analytical method description and lipids, and I have no further comments.